# Ego-Foresight: Self-supervised Agent Visuomotor Prediction for Efficient RL

## Abstract

Despite the significant advancements in Deep Reinforcement Learning (RL) observed in the last decade, the amount of training experience necessary to learn effective policies remains one of the primary concerns both in simulated and real environments. Looking to solve this issue, previous work has shown that improved training efficiency can be achieved by separately modeling agent and environment, but usually requiring a supervisory agent mask. In contrast to RL, humans can perfect a new skill from a very small number of trials and in most cases do so without a supervisory signal, making neuroscientific studies of human development a valuable source of inspiration for RL. In particular, we explore the idea of motor prediction, which states that humans develop an internal model of themselves and of the consequences that their motor commands have on the immediate sensory inputs. Our insight is that the movement of the agent provides a cue that allows the duality between agent and environment to be learned. To instantiate this idea, we present Ego-Foresight, a self-supervised method for disentangling agent and environment based on motion and prediction. Our main finding is that visuomotor prediction of the agent provides good feature representations for the underlying RL algorithm. To test our approach, we integrate Ego-Foresight with a model-free RL algorithm to solve simulated robotic manipulation tasks, showing its ability to improve efficiency and performance in different tasks while making strides towards real-world RL applications, by removing the need for costly supervisory signals.

## 1 Introduction

While it usually goes unnoticed as we go about our daily lives, the human brain is constantly engaged in predicting imminent future sensori inputs (Clark, 2015). This happens when we react to a friend extending their arm for a handshake, when we notice a missing note in our favorite song, and when we perceive movement in optical illusions (Watanabe et al., 2018). At a more fundamental level, this process of predicting our sensations is seen by some as the driving force behind perception, action, and learning (Friston, 2010). But while predicting external phenomena is a daunting task for a brain, motor prediction (Wolpert & Flanagan, 2001) - i.e. predicting the sensori consequences of one's own movement - is remarkably more simple, yet equally important. Anyone who has ever tried to self-tickle has experienced the brain in its predictive endeavors. In trying to predict external (and more critical) inputs, the brain is thought to suppress self-generated sensations, to increase the saliency of those coming from outside - making it hard to feel self-tickling (Blakemore et al., 2000).

In artificial systems, a significant effort has been devoted to learning World Models (Ha & Schmidhuber (2018); Finn et al. (2016); Gumbsch et al. (2024)), which are designed to predict future states of the whole environment and allow planning in the latent space. Despite encouraging deployments of these models in real-world robotic learning (Wu et al., 2023), their application remains constrained to safe and simplified workspaces, with sample-efficient Deep being one of the main challenges.

Though comparatively less explored, the idea of separately modeling the agent and the environment has also been investigated in RL, with previous work demonstrating improved sample-efficiency in simulated robotic manipulation tasks (Gmelin et al., 2023). Additionally, this type of approach has been used to allow zero-shot policy transfer between different robots (Hu et al., 2022) and to improve

environment exploration (Mendonca et al., 2023). Common to all these works is the reliance on supervision to obtain information about the appearance of the robot, allowing to explicitly disentangle agent from environment. This is usually provided in the form of a mask of the agent within the scene, which is obtained either from geometric IDs in simulation, by fine-tuning a segmentation model or even by resorting to the CAD model of the robot.

As humans, we don't receive such a detailed and hard to obtain supervisory signal and yet, during our development, we build a representation of ourselves (Watson, 1966) capable of adapting both slowly, as we grow, and

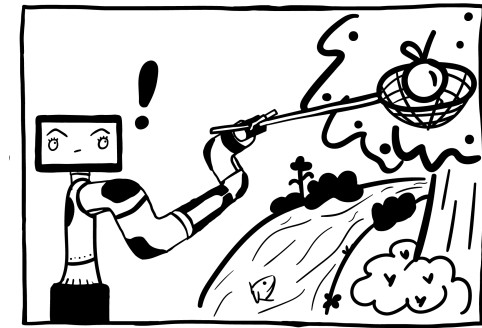

Figure 1: Aware that its embodiment will not allow her to pick up the hanging fruit, a robot uses a tool to extend its reach.

quickly, when we pick up tools (Maravita & Iriki, 2004). In this work, we argue that self-supervised awareness of *self* can also be achieved in artificial systems, and study its advantages relative to supervised methods.

In our approach, which we name Ego-Foresight (EF), in alternative to some recent lines of research (Padalkar et al., 2023), we place the agent's embodiment as an intrinsic part of the learning process, since it determines the visuomotor sensations that the agent can expect as it moves. Our insight is that agent-environment disentanglement can be achieved by having the agent move while trying to predict the visual changes to its body configuration; and that awareness and prediction of the agent's movements should improve its ability to solve complex tasks, as exemplified in Figure 1.

To implement this, we use an encoder-decoder model, that receives as input a limited amount of context RGB frames and the future sequence of proprioceptive states (sense of self-position, e.g. joint angles) that will result from the planned action sequence. The decoder outputs the RGB reconstruction corresponding to the proprioceptive signal, with the agent in its future configuration. This framework naturally lends itself to self-supervised training, by having the system predict future frames.

To test the quality of the learned representations, we combine our architecture with a model-free RL algorithm and assess its performance on multiple simulated domains and on different robotic manipulation and locomotion tasks, demonstrating that our approach can improve sample efficiency and performance. We consider that the factors contributing to this result that the disentanglement between agent and environment allows the RL algorithm to focus its capacity on learning the control of the agent in the initial stages of training, and later on the external aspects and potential interactions within the environment. Our approach combines concepts of model-based RL, in learning a model of the agent, while i) avoiding the need to learn a transition function, and ii) requiring prediction only at training time, for feature learning. In summary, the key contributions of this paper are the following:

- We propose a self-supervised method for disentangling agent and environment based on motion and self-prediction, removing the supervision required by previous methods.

- We integrate our model with a model-free RL algorithm, showing the ability of the learned feature to improve sample-efficiency and performance, in simulated robotic manipulation and locomotion tasks.[1]

- We study the impact of the different components of our approach by performing multiple ablations studies.

## 2 RELATED WORK AND BACKGROUND

**Learning agent representations** The notion of distinguishing self-generated sensations from those caused by external factors has been studied and referred to under different terms, and with a broad range of applications as motivation. Originating in psychology and neuroscience, with the study of contingency awareness (Watson, 1966) and of sensorimotor learning (motor prediction) (Wolpert

---

[1]Code available at: https://github.com/e4s8/ego-foresight.

et al. (2011); Wolpert & Flanagan (2001)), in the last few years this concept has seen growing interest in AI, as an auxiliary mechanism for learning.

In developmental robotics, Zhang & Nagai (2018) have focused on this problem from the standpoint of self-other distinction, by employing 8 NAO robots observing each other executing a set of motion primitives, and trying to differentiate *self* from the learned representations. Lanillos et al. (2020) note that to answer the question "Is this my body?", an agent should first learn to answer "Am I generating those effects in the world". Their robot learns the expected changes in the visual field as it moves in front of a mirror or in front of a twin robot and classifies whether it is looking at itself or not. Our approach is somewhat analogous to these works, in the sense that we identify as being part of the agent that which can be visually predicted from the future proprioception states, while the robot moves. In a related, but reverse direction, Wilkins & Stathis (2023) propose the act of doing nothing as a means to distinguish self-generated from externally-generated sensations.

Still in robotics, the idea of modeling the agent has connections with work in body perception and visual imagination for goal-driven behaviour (Sancaktar et al., 2020) as well as self-recognition by discovery of controllable points (Edsinger & Kemp (2006); Yang et al. (2020)). Another application that has been explored is the learning of modular dynamic models, that decouple robot dynamics from world dynamics, allowing the latter to be reused between robots with different morphologies. Hu et al. (2022) propose a method for zero-shot policy transfer between robots, by taking advantage of robot-specific information - such as the CAD model - to obtain a robot mask from which future robot states can be predicted, given its dynamics. These future states are then used solve manipulation tasks using model-based RL. Finally, this concept has also been used for ignoring changes in the robot as a means for measuring environment change, with the intention of incentivizing exploration in real household environments (Mendonca et al., 2023).

In machine learning (ML), the distinction between the agent and environment has been studied under the umbrella of disentangled representations, a long-standing problem in ML (Bengio, 2013). While most works take an information theoretic approach to disentangled representation learning (Higgins et al. (2017); Kim & Mnih (2018); Chen et al. (2016)), some try to take advantage of known structural biases in the data, which is particularly relevant for sequential data, as it usually contains both time-variant and invariant features (Wiskott & Sejnowski, 2002). In video, this allows the disentanglement of content and motion (Villegas et al., 2017). Denton & Birodkar (2017) explore the insight that some factors are mostly constant throughout a video, while others remain consistent between videos but can change over time to disentangle content and pose. The scene encoder of our method is based on the model of Denton and Birodkar, but while they use an adversarial loss to model pose as a property that doesn't depend on the video, we take advantage of the notion of agency and use the proprioceptive signal to model the visuomotor mapping with a simple reconstruction loss.

Finally, agent-environment disentanglement can also be achieved through attention mechanisms (Choi et al., 2018) or from explicit supervision, as shown by Gmelin et al. (2023) with SEAR. In this work, the authors demonstrate that learning this distinction using a supervisory mask of the agent allows RL algorithms to achieve better sample-efficiency and performance, serving as the most direct baseline for our work. Still, while in simulation this mask may be inexpensive to obtain, in real world scenarios it requires training a dedicated segmentation model and manually labeling ground truth images for supervision, for each new robot that is tested, reducing the real-world applicability of SEAR. One final related line of research is the learning of representations that are invariant to task-irrelevant information, either via Bisimulation Zhang et al. (2021) or by denoising distractors using an explicit factorization of the representation (Fu et al. (2021); Wang et al. (2022)).

**RL from Images** RL problems are typically formulated as Markov Decision Problems, defined as a tuple $(\mathcal{S}, \mathcal{A}, \mathcal{T}, \mathcal{R}, \gamma, d_0)$, where $\mathcal{S}$ is the state space, $\mathcal{A}$ is the action space, $\mathcal{T}(\boldsymbol{s}_{t+1}|\boldsymbol{s}_t, \boldsymbol{a}_t)$ is the transition function, $\mathcal{R}(\boldsymbol{s}_t, \boldsymbol{a}_t)$ is the reward function, $\gamma \in [0, 1]$ is a discount factor and $d_0$ is the distribution over initial states $\boldsymbol{s}_0$. The objective in RL is to learn the policy $\pi : \mathcal{S} \to \mathcal{A}$ that maximizes the expected discounted cumulative reward $\mathbb{E}_\pi[\sum_{t=0}^{\infty} \gamma^t \mathcal{R}(\boldsymbol{s}_t, \boldsymbol{a}_t)]$, with $\boldsymbol{a}_t \sim \pi(\cdot|\boldsymbol{s}_t)$ and $\boldsymbol{s}_{t+1} \sim \mathcal{T}(\cdot|\boldsymbol{s}_t, \boldsymbol{a}_t)$.

Over the last decade, work in Deep RL from images, where environment representations are learned from high-dimensional inputs, has allowed RL agents to solve problems for which features cannot be designed by experts. Since the first demonstration of human-level performance being achieved on arcade games (Mnih et al., 2013), numerous algorithmic improvements have been introduced.

Of particular relevance to our work is Experience Replay (Mnih et al., 2015) for improved training stability, in which episode steps are stored in a replay buffer, allowing updates to be performed on randomly drawn samples of past experience. Another significant addition was the introduction of augmentation techniques to improve performance and efficiency in model-free RL from images (Yarats et al., 2021b). In this work we adopt the augmentation proposed in Yarats et al. (2021a), implemented using DDPG (Lillicrap et al., 2016).

DDPG is an off-policy actor-critic RL algorithm for continuous control, that alternates between learning an approximator to the Q-function $Q_\phi$ and a deterministic policy $\pi_\theta$. Specifically, we adopt the Twin Delayed variation of DDPG (Fujimoto et al., 2018), which adds clipped douple Q-learning and delayed policy updates to limit over-estimation of the Q-values. Having sampled a batch of transitions $\tau = (s_t, a_t, r_{t:t+n-1}, s_{t+n})$ from the replay buffer $\mathcal{D}$, $Q_\phi$ is learned by minimizing the mean-squared Bellman error:

$$\mathcal{L}_{critic}(\phi, \mathcal{D}) = \mathop{\mathbb{E}}_{\tau \sim \mathcal{D}} \left[ (Q_{\phi_k}(s_t, a_t) - y)^2 \right] \quad k \in \{1, 2\}, \tag{1}$$

using target networks $Q_{\hat{\phi}_k}$ to approximate the target values, with $n$-step returns, and where $\hat{\phi}$ are slowly updated copies of the parameters $\phi$ :

$$y = \sum_{i=0}^{n-1} \gamma^i r_{t+i} + \gamma^n \min_{k=1,2} Q_{\hat{\phi}_k}(s_{t+n}, \pi_\theta(s_{t+n})). \tag{2}$$

The policy is learned by maximizing $\mathbb{E}_{s \sim \mathcal{D}} \left[ Q_\phi(s_t, \pi_\theta(s_t)) \right]$, to find the optimal action with respect to the Q-function. Even though DDPG has been successfully applied to RL from images, its instability means that results should be reported on runs from multiple random seeds (Islam et al. (2017); Henderson et al. (2018)).

In model-based RL, approaches such as Dreamer (Hafner et al., 2020) have explored the idea of training in an imagined latent space, by sampling thousands of parallel trajectories. Similarly to our work, some approaches such as Self-Predictive Representations (Schwarzer et al., 2020)) and Imagination-Augmented Agents (Racanière et al., 2017) have combined ideas from model-free and model-based RL, by augmenting model-free algorithms with prediction based auxiliary losses. However, these works require learning a full dynamics model, they do not consider any distinction between agent and environment and focus on discrete problems.

**Controllable video generation and prediction** In learning to generate future frames conditioned on the expected proprioceptive states of the robot, our proposed model presents connections with controllable video generation (Menapace et al. (2021; 2022); Davtyan & Favaro (2022)) and action-conditioned video prediction (Oh et al. (2015); Finn et al. (2016); Dehban et al. (2022); Nunes et al. (2020)), opening the door for future applications with model-based RL. These are research areas that have seen growing interest in the past year, with the release of large-scale models such as Genie (Bruce et al., 2024) and UniSim (Yang et al., 2024), which aim at learning simulators for both games and real-world scenes, using large amounts of diverse data. Though comparatively, our model is significantly lighter, we believe the design principle of predicting the agent could be scaled to generalize to more complex or diverse environments, provided that the architecture is adapted with higher capacity models. In appendix A.1 We show experiments on the BAIR dataset, with generation of previously unseen sequences on a real-world environment.

## 3 APPROACH

### 3.1 EPISODE PARTITION

In a real-world scenario, in which the agent is a robot, it would be reasonable to assume that there is access to a camera video stream and, for each planned action sequence, the sequence of future expected proprioceptive states in the form of joint angles and Cartesian position of the end-effector, obtained from forward kinematics. Hence, we define that our dataset is composed of $N$ episodes $\tau_i$ of a fixed length $L$ (Figure 2), which include a sequence of states $s$, composed of RGB frames $x$ and

the corresponding proprioceptive states $\boldsymbol{p}$ of the agent $\boldsymbol{\tau}_i = \{\boldsymbol{s}_0, ..., \boldsymbol{s}_L\}_i = \{(\boldsymbol{x}_0, \boldsymbol{p}_0), ..., (\boldsymbol{x}_L, \boldsymbol{p}_L)\}_i$, $i = 1, ..., N$. During training, we randomly select a window of size $C + H$ within each episode - which corresponds to the number of context frames plus the prediction horizon, respectively. This artificially augments the available data, by creating different observations within each episode. Finally, from each window, a fixed number $C$ of context time steps at the start are taken for input to the model,

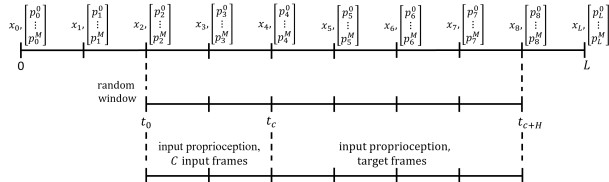

Figure 2: Partition of an episode $\boldsymbol{\tau}_i$. From each sequence of frames and proprioception states (top), a window of size $C + H$ is randomly sampled (middle). The first $C$ steps of the window are used as context. For the remaining steps, the proprioception states are used as input, while the frames serve as reconstruction target.

as well as the proprioceptive states for the whole sequence up to $H$. The RGB frames between $t_C$ and $H$ are used as target for the prediction.

## 3.2 MOTION-BASED AGENT-ENVIRONMENT DISENTANGLEMENT

To model future visual configurations of the robot, we propose an encoder-decoder model (Figure 3) that has two encoder streams and one decoder head. The first encoder, parameterized by $\psi$ produces a representation for the visual content of the scene $\boldsymbol{h}_s^{t_c} = E_{s_\psi}(\boldsymbol{x}^{t_0:t_c})$, obtained from the context frames. The second encoder, parameterized by $\mu$, creates a feature representation for the sequence of proprioceptive states $\boldsymbol{p}$ of the agent $\boldsymbol{h}_p^{t_n} = E_{p_\mu}(\boldsymbol{p}^{t_0:t_n})$ and is implemented with an LSTM. These two representations are concatenated and decoded into the future frame observation $\hat{\boldsymbol{x}}^{t_n} = D_\zeta(\boldsymbol{h}_s^{t_c}, \boldsymbol{h}_p^{t_n})$, with $\zeta$ being the parameters of the decoder and $n \in \{C, ..., H\}$ a randomly chosen timestep. This formulation results in the following reconstruction loss term:

$$\mathcal{L}_{rec}(\psi, \mu, \zeta, \boldsymbol{x}^{t_0:t_c}, \boldsymbol{x}^{t_n}) = ||\hat{\boldsymbol{x}}^{t_n} - \boldsymbol{x}^{t_n}||_2^2, \tag{3}$$

To ensure that the scene representation is independent from the timestep and doesn't contain dynamic information from the agent's motion, we add a second objective based on the assumption that, for the most part, the scene content varies slowly over time, remaining mostly the same during the same episode, but changing from episode to episode. Hence, we adopt the similarity loss from Denton & Birodkar (2017), to encourage the distance between scene representations of frames coming from the same episode to be small:

$$\mathcal{L}_{sim}(\psi, \boldsymbol{x}^{t_0:t_c}, \boldsymbol{x}^{t_0+m:t_c+m}) = ||E_s(\boldsymbol{x}^{t_0:t_c}) - E_s(\boldsymbol{x}^{t_0+m:t_c+m})||_2^2 = ||\boldsymbol{h}_s^{t_c} - \boldsymbol{h}_s^{t_c+m}||_2^2, \tag{4}$$

where $m$ is a time delta chosen at random. The complete training objective is then the expected value over a batch of sequences sampled from a dataset $\mathcal{D}$:

$$\mathcal{L}_{ef}(\psi, \mu, \zeta, \mathcal{D}) = \mathbb{E}_{\boldsymbol{x}^{t_0:t_c+H}, \boldsymbol{p}^{t_0:t_c+H} \sim \mathcal{D}}[\mathcal{L}_{rec}(\psi, \mu, \zeta) + \alpha \mathcal{L}_{sim}(\psi)], \tag{5}$$

where $\alpha$ controls the weight of the similarity loss term, and should be adjusted according to how much the content of a scene is expected to change, allowing the initial assumption to be relaxed.

## 3.3 AGENT VISUOMOTOR PREDICTION AS FEATURE LEARNING FOR RL

To test how our model affects sample-efficiency in RL, we implement it together with DDPG. The episodes stored in the replay buffer $\mathcal{D}$ allow us to maintain the approach described in section 3.2 while jointly training our model and learning the policy.

Similarly to Gmelin et al. (2023), the Actor and Critic neural networks of DDPG receive the low dimensional feature vector coming from the encoders, which is the concatenation of $\boldsymbol{h}_s$ and $\boldsymbol{h}_p$ at $t = t_c$. To jointly train our model and the RL algorithm, we optimize our model together with the critic loss of equation 1, resulting in the final objective function $\mathcal{L}$ that is the total of both terms:

$$\mathcal{L}(\phi, \psi, \mu, \zeta, \mathcal{D}) = \mathcal{L}_{critic}(\phi, \mathcal{D}) + \mathcal{L}_{ef}(\psi, \mu, \zeta, \mathcal{D}). \tag{6}$$

Typically, training DDPG involves first sampling an episode from the replay buffer, and then sampling one transition of that episode for the update of the actor-critic. Our approach requires two extra

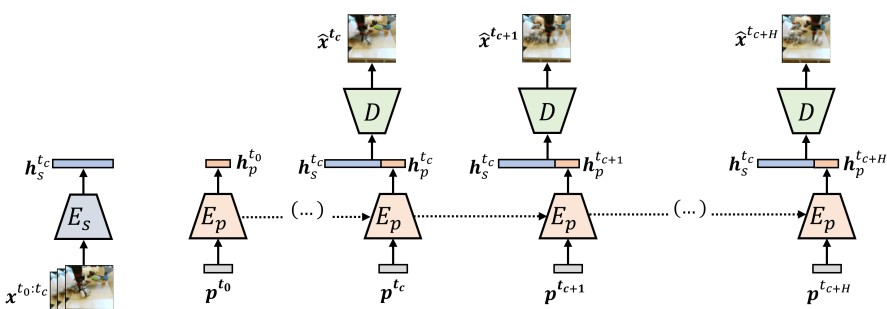

Figure 3: Visuomotor prediction using Ego-Foresight.

random choices from the same episode: first the sequence of transitions used for frame prediction and computing $L_{rec}$, and second the $C$ transitions used to obtain the similarity loss target $h_s^{t_c+m}$.

As we developed our method, we noticed that, when training our model together with the RL algorithm, the agent tended to perform goal-directed movements, seeking to maximize reward which prevented our model from observing diverse enough motions to learn the visuomotor mapping. To solve this issue we introduced a motor-babbling (Saegusa et al. (2009); Kase et al. (2021)) stage for a fixed number of steps at the start of training, during which actions are random choices of $\pm 1$, forcing exploratory movements.

It is important to note however, that by jointly training the RL algorithm and our model, the visual quality of the reconstructed frames is still reduced when compared to that achieved for the BAIR dataset (A.1), for which EF is trained without DDPG, purely on prediction. However, in improving performance and sample-efficiency in RL tasks, the reconstruction of future frames serves as an auxiliary objective, which can be relaxed, as long as both the ability to model the effect of the commanded actions on the configuration of the agent is preserved.

## 4 EXPERIMENTS & RESULTS

### 4.1 EXPERIMENTAL SETUP

**Environments**  We test our approach in 3 different environments, starting with the DeepMind Control Suite (DMC) (Tassa et al., 2018), a widely used benchmark for continuous control. Set in an empty and open world, it consists of a range of physical control tasks, such as locomotion, involving a variety of agents, with different body configurations and number of degrees of freedom, allowing us to test the generality of our approach to different embodiments and robots. The second environment we test is the Distacting Control Suite (Stone et al., 2021), a variant of DMC which replaces the background of each sequence with a randomly chosen image, allowing a better assessment of the robustness of RL algorithms to the visual complexity of the real world, something the authors demonstrate is a limitation of current methods. Finally, Meta-world is a simulated benchmark (Yu et al., 2020) of robotic manipulation tasks with predefined reward functions. It presents a broad distribution of tasks which require a diverse set of skills such as reaching, picking, placing, or inserting. We chose 10 that involve the manipulation of different objects. More details can be found in App. A.5. All environments are implemented using the Mujoco simulator (Todorov et al., 2012).

**Baselines**  We compare our model with three baselines: DrQ-v2 (Yarats et al., 2021a), SEAR (Gmelin et al., 2023), Dreamer-v3 (Hafner et al., 2023) and TD-MPC2 (Hansen et al., 2024), introduced in section 2. DrQ-v2 uses an encoder to embed the RGB frames into a feature vector which models the full environment. SEAR expands on top of DrQ-v2, by splitting the feature representation into agent and full environment information, which is achieved with a supervisory mask. SEAR is therefore the closest baseline to our model and, for its use of supervision, can be viewed as an oracle baseline when compared to ours. In all experiments bellow, the supervisory mask is obtained directly from the simulator. Finally, Dreamer-v3 is a model-based RL algorithm which aims to improve sample-efficiency by learning from outcomes of actions which are imagined using a world model. The results of DrQ-v2 on DMC, and Dreamer-v3 and TD-MPC2 on DMC and

Meta-World are provided in tables, available at each project's webpage. All remaining curves are obtained using the source code provided by the authors of each paper. We implement our model as an extension to the code of Gmelin et al. (2023), which in turn is based on the source code of Yarats et al. (2021a). For SEAR, despite using the publicly available code, we failed to match the results reported in the original paper. We note however, some inconsistencies in the original plots such as success rates above 1. We perform multiple runs for each experiment using different random seeds however, due to time constraints, not all baselines and experiments use the same number of random seeds. We specify the number of random seeds in A.4.

**Architecture** In our experiments, we chose an implementation of the scene encoder $E_s$ and the decoder $D$ based on the DCGAN (Radford et al., 2016), which consists of 5 convolutional layers, each reducing (in $E_s$) or increasing (in $D$) the size of the feature map by half. To provide a sequence of context frames as input, we stack the frames along the channels dimension. Another change relative to DCGAN is including skip connections from the $E_s$ to the $D$, as done in the U-Net (Ronneberger et al., 2015). For the proprioception encoder, we use an LSTM preceded by 2 fully connected layers. More details can be found in A.3. We note that our choice of architecture is determined by the low resolution of the RGB observations in the tested environments, which is at maximum $84 \times 84$ pixels. Nevertheless, higher capacity architectures could be used to deal with higher dimensional inputs, provided enough computational resources. We chose most hyperparameters to match those used in SEAR (see A.5), including the size of the feature vectors. Additionally, for a fair comparison to the baselines, we perform the motor-babbling stage (50k steps) from scratch for each task. In practice however, it would suffice to go through this stage once per robot body (e.g. Meta-world's Sawyer) and then start learning the different tasks from a model pre-trained on babbling.

**Metrics** To compare our model against the baselines, we plot the mean episode reward for the DMC and Distracting DMC benchmarks and the mean success rate for Metaworld, both as a function of environment steps. Environment steps represent the number of times the environment is updated according to an action. Because environment updates incur a significant computational cost, they are the preferred way of reporting sample-efficiency in the literature (Yarats et al. (2021a), Hafner et al. (2020)). Each curve presents the mean over 10 evaluation episodes, taken every 5k steps. Furthermore, in trying to reduce the instability of DDPG, we also average the performance over multiple random seeds and display the standard deviation in the shaded area of the plot.

## 4.2 PHYSICAL CONTROL AND LOCOMOTION EXPERIMENTS

**DeepMind Control Suite** We test our model on 4 different tasks of the DMC Suite (Figure 4), using the Cartpole, Walker, Cheetah and Quadruped agents, which have 2, 7, 8 and 16 degrees of freedom respectively. For every task we match or improve the performance of the baselines. We note that the greater variability in our results is due to a lower number repetitions with different random seeds. The results of EF in terms of sample-efficiency are also competitive with the baselines, with the exception of the Walker task. Finally because during the babbling stage the agent is not acting towards the maximization of the reward, the first 50k steps of the Cartpole task perform worse than the baselines but, once this stage is over, the agent quickly solves the task. A noteworthy aspect of this experiment is the fact that the greater the complexity of the agent in terms of degrees of freedom, the more advantage our approach offers, with the Quadruped experiment being the one in which it achieves better relative performance, supporting the claim that self-prediction can facilitate control of the agent's embodiment.

**Distracting Control Suite** On the Distracting DMC benchmark we test 2 tasks (Figure 5), in which we once again match the results of the baseline without resorting to a supervisory GT mask for learning the agent-environment separation. It can be observed that in the Cheetah task, the performance is lower than the one achieved in DMC, but still enough for the agent to learn how to run. In terms of efficiency, our approach outperforms the baseline in the Cheetah task and matches it on Cartpole. Success in this benchmark - in which it would be easy for the model to get stuck trying to predict and model the changing visual environment - helps support our claim that our approach achieves an understanding of self, which allows it to still control the agent despite the added distractions. Still, the degradation in performance on the Cheetah task suggests that the agent-environment decoupling is imperfect and there is still progress to be made.

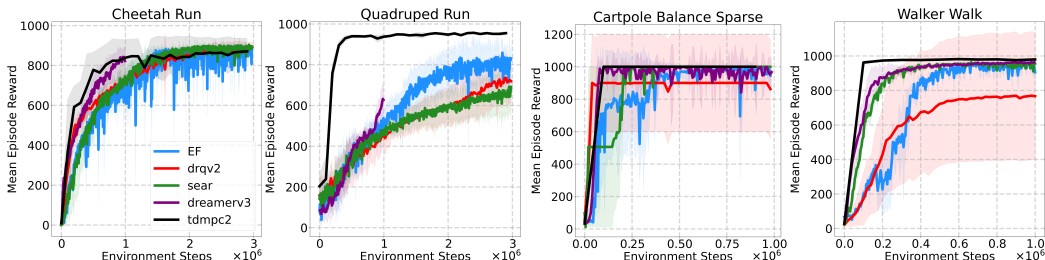

Figure 4: Mean episode reward on 4 tasks of the DeepMind Control Suite.

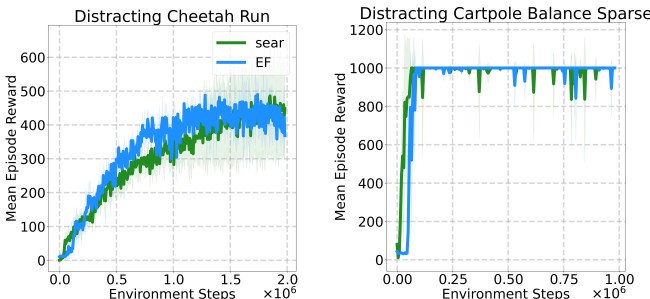

Figure 5: Mean episode reward on 2 tasks of the Distracting Control Suite.

### 4.3 SIMULATED ROBOT MANIPULATION TASKS

**Quantitative Results**    Besides the physical control tasks tested in the DMC and Distracting benchmarks, we investigate the ability of our approach to learn representations that allow interaction and manipulation of objects present in the environment by testing 10 different Meta-world tasks. As in the previous experiments, we analyse the Meta-world results, presented in Figure 6 in terms of performance and sample-efficiency. We observe that EF plainly achieves the best success rate in 3 out of the 10 tasks, matches the baselines in 3 others and under performs at least one baseline in the 4 remaining tasks. Regarding sample-efficiency, our model is the most efficient in 4 of 10 tasks, but also the least efficient in 1 task. One drawback of our method when it comes to efficiency, is the need for the motor-babbling stage, which sets a lower bound on how fast the agent can learn. For example, in the Door Close and Button Press Wall tasks, our method starts being successful immediately after the 50k babbling steps, hinting that in those cases this stage could have been shorter. In the future, this could be improved to make babbling stop once the visuomotor mapping has been learned.

When analyzing the results of Figure 6, it is important to consider two other factors: supervision and hyperparameter fine-tuning. The three tested models differ in how they are trained, with SEAR needing supervision to disentangle the agent, and DrQ-v2 not requiring supervision, but modeling the complete environment in the feature vector. Our model combines both characteristics, extending its range of applications. In terms of hyperparameters, and due to computation and time constraints, we use the same set of hyperparameters on all our experiments. We believe that a more careful choice could further improve the results.

**Qualitative Results**    In Figure 7, we highlight two of the tasks, showing the sequence of actions taken by the agent to solve the tasks and how, for those action sequences, the motion of the agent is very well predicted by EF. In particular, we note how for the Door Open task, the movement of the arm is predicted while the door is kept static as part of the scene, denoting the model's understanding of what constitutes the agent and what is part of the scene. This contrasts with the Hammer task, in which besides the movement of the agent, the model also predicts how the hammer moves. This is because once the hammer is picked up, it effectively is integrated into the robot's body, and can therefore be predicted from the future proprioceptive states.

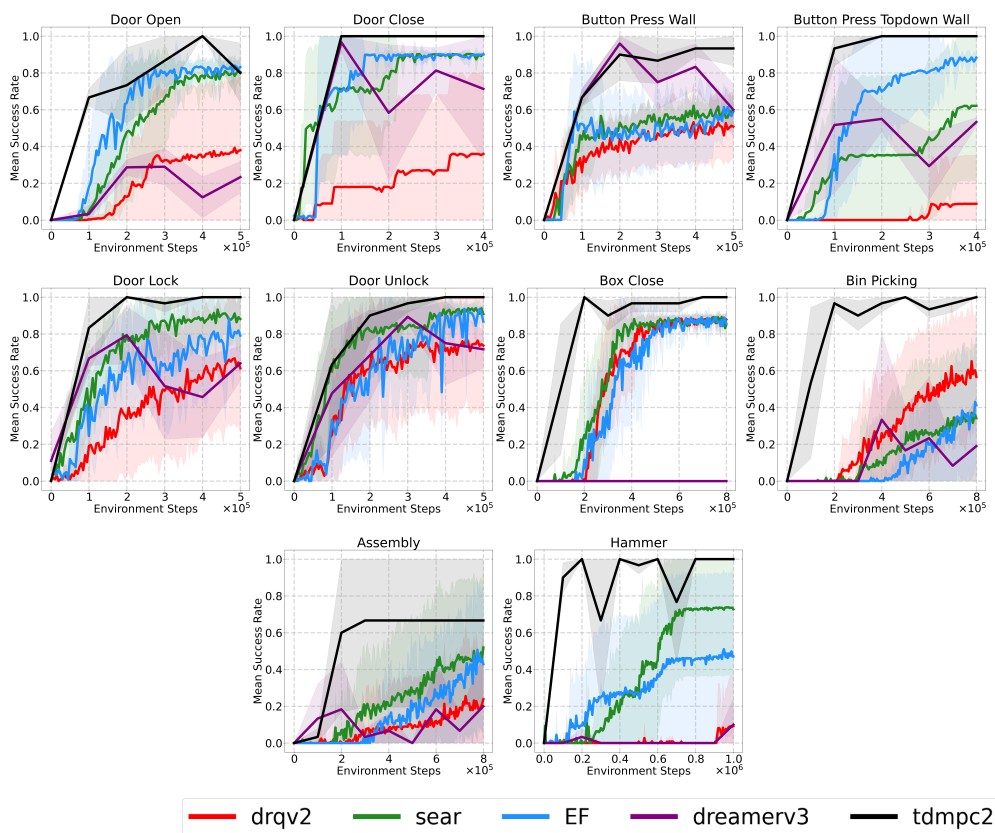

Figure 6: Mean success rate for on 10 different Meta-world tasks.

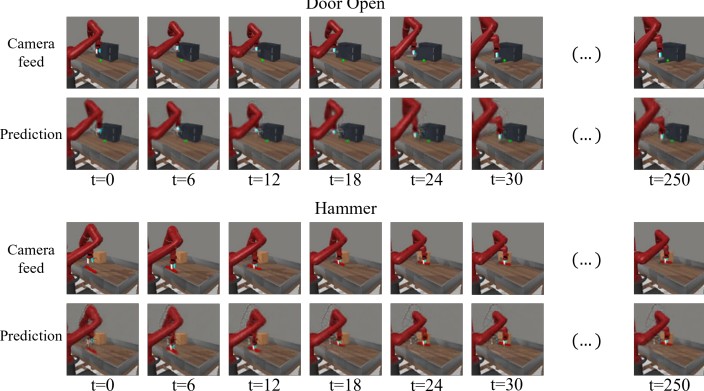

Figure 7: The policy's solution to two tasks and the model prediction for the sequence of states that solves the task.

### 4.4 Ablation Tests

To study how the parameters introduced by our method influence performance and efficiency, we run EF on a fixed task, for different values of $H$ and $\alpha$. Figure 8 shows the effect of varying the prediction horizon, indicating that longer prediction horizons can lead to better efficiency and performance. It is important to note however, that longer horizons incur a greater computational cost at training time, since the LSTM must back-propagate the gradient through the whole sequence. As such, we do not explore prediction horizons longer than 40.

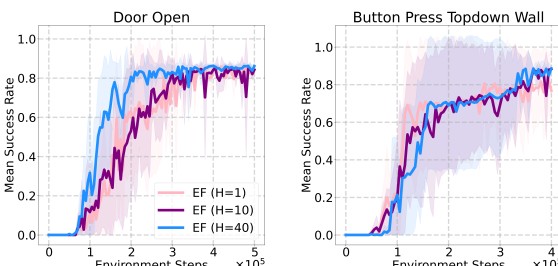

Figure 8: Ablation study of the prediction horizon $H$.

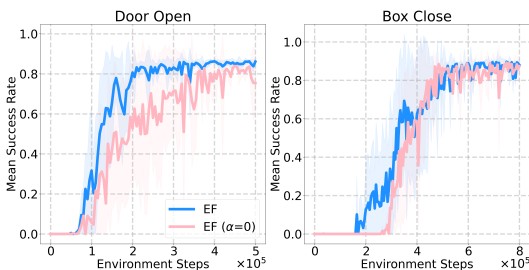

Figure 9: Ablation study of the similarity loss.

To test the effect of the similarity loss on the success rate in downstream tasks, we set $\alpha = 0$ and report the results on two different Meta-world tasks (Figure 9). We find that the effect of the similarity loss is stronger in tasks in which big variations in the scene occur, such as the Door Open task. For tasks in which visual variations are smaller variations, or mostly due to objects that are integrated into the body of the robot (e.g. Box-Close or Hammer) and can thus be well predicted, the effect of the similarity loss is less noticeable. Further ablation tests, related to the proprioceptive input of EF, are provided in Appendix A.2.

## 5 DISCUSSION AND FUTURE WORK

In this work, we studied how motion can be used to disentangle agent and environment in a self-supervised manner. We integrate our approach with an RL algorithm and evaluate the effect of the learned feature representations on RL physical control and manipulation tasks solved in simulation. Our results show that in most tasks, our approach matches the performance of the best baselines, and sometimes achieves better results. By removing the need for supervision while preserving sample-efficiency, our method can be seen as a step towards model-free RL in real-world settings. Some limitations of our work include the assumption that there are no big changes in the environment inside the prediction horizon and the need for pixel-wise reconstruction of the scene, for which different approaches such as a contrastive loss could be explored in the future. The fixed babbling stage is another limitation, which should be made adaptive. Still, it opens up the possibility of reusing a model pre-trained on babbling to improve sample-efficiency on multi-task learning. Finally, our approach is comparable to SEAR in terms of wall-clock efficiency, but more expensive than DrQ-v2. However, we consider that for future work aiming at training RL methods directly on real-world robots, the main bottleneck lies in the number of episodes that can be obtained, whereas the learning algorithm only needs to run at least at the frequency of the robot's control loop. For this reason we focus on sample-efficiency. Our results also indicate that one benefit of self-supervision is adaption to a changing body-schema, in particular when the agent uses tools. In future research, we intend to explore how fast this body-schema adaptation can be done, and whether performance on tasks that require tools can benefit from it. Finally, we intend to apply our method to domains outside robotics, such as autonomous driving, where the main difference is that the actions of the agent control the optical flow of the observed world and not the configuration of the agent's body. Here, our method can additionally be used as a regularizing loss term for encouraging action that lead to predictable states, a beneficial feature in automated driving.

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

## A    APPENDIX

### A.1    AGENT-ENVIRONMENT DISENTANGLEMENT IN REAL-WORLD DATASET

**BAIR Dataset**    Commonly used in video prediction research, the BAIR dataset (Ebert et al., 2017) consists of a Sawyer robot arm randomly moving and pushing a broad range of objects inside a confined table. The set of objects and their disposition changes from video to video, as well as the lighting conditions, providing some variability. It includes 43520 video samples, each 30 frames long, with $64 \times 64$ resolution. each sequence includes the robot's actions, defined as a 7D array of Cartesian displacement and position of the end-effector and the gripper open/close state. The model is trained by receiving 2 context frames and predicting the following 10. At test time, the prediction horizon is extended to 28, with the context length staying the same. The proprioception is defined as the concatenation of the robot's action and the gripper state.

**Results**    In Figure 10, we show two sample predictions from EF on the BAIR dataset. The model succeeds in separating visual information that is part of the robot from the scene, predicting the trajectory of the robot's arm according to its true motion, which shows that the model correctly learned the visuomotor map between proprioception and vision. The background, including moving objects, is reconstructed in their original position. Even so, our model still predicts that some change should happen when the arm passes by an object, and it often blurs objects that predictably would have moved. Nevertheless, it should be noted that the agent's actions and object movement are inherently correlated and therefore can't be completely disentangled.

To determine whether EF could be adapted to be used with model-based planning algorithms - that work by imagining the expected outcome of multiple different trajectories in parallel - we evaluate its ability to generalize to previously unseen trajectories. To achieve this, we train EF on BAIR dataset using only the Cartesian position of the gripper as proprioception. This allows us to handcraft artificial and previously unseen movements, as shown in Figure 11. While we display a single handcrafted example, this simple experiment shows that, provided a policy function, the dynamics learned by EF should manage to predict the outcome of sampled trajectories.

### A.2    ADDITIONAL ABLATIONS

We report two additional ablations (Figures 12 and 13), in addition to the ones presented in section 4.4. First, we investigate the reliance of our method on the availability of the joint state information in the Meta-world experiments. To do this, we train a variation of EF that only has access to the Cartesian position of the robot's end-effect (the commanded actions) and compare the results to our full model (see Figure 12). This removes the need for extra sensors, compared to baselines. We find that the Cartesian position as proprioceptive state suffices for the agent to learn how to solve the tasks, with the addition of joint information even hindering the results in tasks such as Door Lock. We attribute this to the fact that in order to solve the presented tasks, it is enough for the RL agent to be aware of the end-effector's position and that the additional joint information can add noise to the representation.

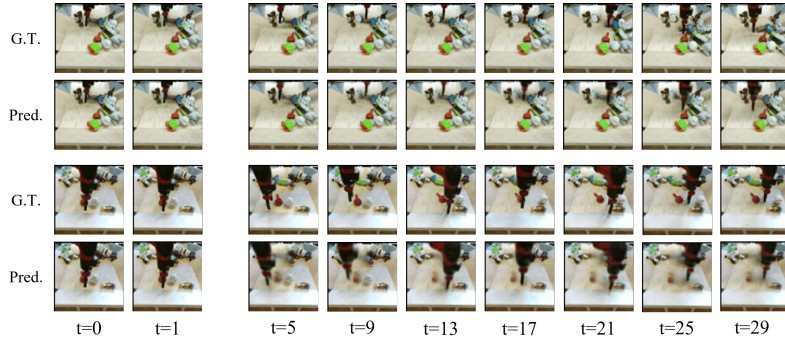

Figure 10:    Predictions on the BAIR Dataset.    See `https://github.com/e4s8/ego-foresight`.

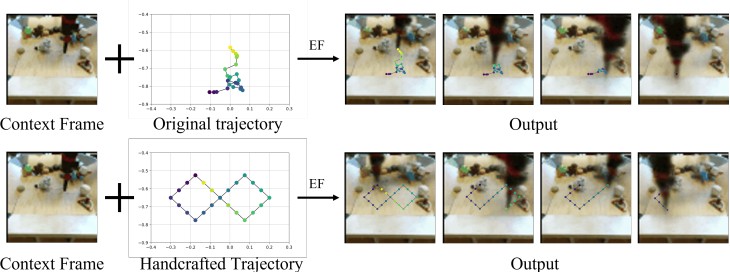

Figure 11: Generation of an unseen trajectory. See `https://github.com/e4s8/ego-foresight`.

Still, in more complex tasks with more constraints on the configuration of the agent, it could be necessary to provide the joint states.

The second ablation investigates whether the addition of agent state information as input is the most significant factor in the results achieved by EF. To exclude this possibility, we implemented versions of DrQ-v2 and SEAR that also receive agent state as input. To do so, we use the same architecture of EF's proprioception encoder to encode the state and obtain $h_p$. For DrQ-v2, $h_p$ is directly concatenated with the already existing RGB feature representation, as we do for EF. In the case of SEAR, the RGB feature representation is decoupled in 2 vectors, so we concatenate $h_p$ to: 1) $z_1$, for the mask decoder; 2) $z_2$, for the full RGB decoder and 3) to $z_1$ and $z_2$, which is used as input the to actor-critic. In this way, we can provide a fair comparison in terms of observation, with the only difference being that the baselines only observe 3 past and current states (since these algorithms do not consider prediction), while EF also makes use of future states for prediction. The results, presented in Figure 13 show that even with the added proprioception state input, EF still remains competitive or improves upon the baselines.

### A.3 DETAILED ARCHITECTURE

The scene encoder $E_s$ is a convolutional neural network with 5 layers. On all layers, we use kernel size 4 and stride 2, except for layer 5 which has stride 1. Each layer uses Batch Normalization and leaky ReLU activations. The number of filters in each layer is fixed at 32. The last layer has number of filters corresponding to the size of $h_s$ and a hyperbolic tangent activation. The decoder is a mirror of the encoder, with input size corresponding to the sum of the sizes of $h_s$ and $h_s$. Additionally, skip connections are added between each Conv. layer of $E_s$ and the corresponding layer of the decoder. The output activation of the decoder is a sigmoid function. For the proprioception encoder we use a two layer fully connected neural network with leaky ReLU activations, followed by an LSTM, each with 256 units. After the LSTM and additional fully connected layer maps the size of the LSTM's output to the size of $h_p$. The size of the feature vectors is chosen to be close to the baselines. Finally, the list of hyperparameters used in the DMC, Distracting Control and Meta-world experiments is presented in Table 2.

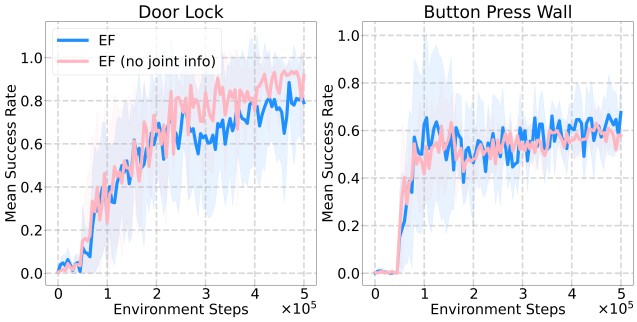

Figure 12: Ablation study of the reliance on joint state information.

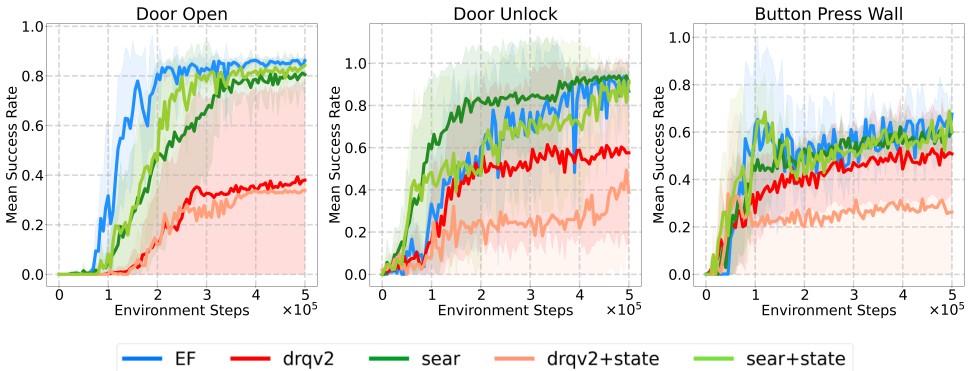

Figure 13: Baselines augmented with state information

## A.4 TRAINING DETAILS

Training was split between one NVIDIA RTX3060, one RTX2070 and one A100. For consistency, all runs for a given task are done in the same machine, varying the algorithm and the random seeds. Depending on the machine, on the prediction horizon and other hyperparameters, each training run takes approximately 2 hours for the BAIR dataset experiments, 4 to 8 hours in the Meta-world experiments and over one day on the DMC and Distracting Control tasks with 3M steps. In terms of VRAM - and for the chosen batch sizes - BAIR dataset experiments require approximately 11GB, and the DMC, Distracting Control and Meta-world experiments require 2.6GB. Finally, due to time constraints and to results for some of the baselines being obtained from their official website, as detailed in section 4.1 not all experiments in our results present the same number of random trials. Nevertheless, we choose to present all available trials for each experiment. We thus present the number of random seeds per experiment in Table 1.

Table 1: Number of random seeds per experiment.

| Experiment | DrQ-v2 | SEAR | Dreamer-v3 | TD-MPC2 | EF |
|---|---|---|---|---|---|
| DMC (Fig. 4) | 10 | 5 | 5 | 10 | 3 |
| Distracting DMC (Fig. 5) | - | 5 | - | - | 3 |
| Metaworld (Fig. 6) | 10 | 10 | 3 | 3 | 10 |
| Horizon ablation (Fig. 8) | - | - | - | - | 5 |
| Similarity ablation (Fig. 9) | - | - | - | - | 5 |
| Joint state ablation (Fig. 12) | - | - | - | - | 5 |
| Baselines without/with prop. (Fig. 13) | 10/5 | 10/5 | - | - | 5 |

## A.5 META-WORLD BENCHMARK DETAILS

Meta-world is a simulated benchmark of robot manipulation tasks, designed for meta-learning and multi-task learning, including more than 50 different tasks. As in BAIR, the agent is a Sawyer robot, but while for BAIR the we use an FPV camera angle, for Meta-world we opt to use a third-person view, as done in (Gmelin et al., 2023). While this means that the number of degrees of freedom that can be observed is higher (the full 9 DoF of the Sawyer), we verify that the model still manages to predict the agent's movement. For the proprioceptive state, we use the 9D array of the robot's joint angles (7 arm joints and 2 fingers) and the 4D end-effector state, which includes the Cartesian position of the gripper and the distance between the fingers. After each for episode the locations of the targets are slightly varied.

Table 2: Default hyperparameters used in the DMC, Distracting Control and Meta-world experiments.

| Shared parameters | |
| --- | --- |
| Parameter | Value |
| Replay buffer size | $2.5 \times 10^5$ |
| Action Repeat | 2 |
| Seed frames | 4000 |
| Exploration steps | 2000 |
| $n$-step returns | 3 |
| Batch size | 256/512 |
| Discount $\gamma$ | 0.99 |
| Optimizer | Adam |
| Learning rate | $1 \times 10^{-4}$ |
| Agent update frequency | 2 |
| Critic Q-function soft-update rate $\tau$ | 0.01 |
| Exploration stddev. clip | 0.3 |
| Exploration stddev. schedule | linear(1.0, 0.1, 500000) |
| Evaluation episodes | 10 |
| Encoder Features dim. | 32 |
| Context frames | 3 |
| Frame size | (84, 84) |
| **DrQ-v2 parameters** | |
| Parameter | Value |
| Hidden dim. | 39200 |
| **SEAR parameters** | |
| Parameter | Value |
| Hidden dim. | 4096 |
| Reconstruction loss weight | 0.01 |
| Mask loss weight | 0.0025 |
| **Ego-Foresight parameters** | |
| Parameter | Value |
| $\boldsymbol{h}_s$ dim. | 4096 |
| $\boldsymbol{h}_p$ dim | 32 |
| Motor-babbling steps | 50k |
| Similarity loss weight $\alpha$ | 0.01 |
| Prediction horizon (time steps) | 40 |

