# OpenReview forum: "Ego-Foresight: Self-supervised Agent Visuomotor Prediction for Efficient RL"
_ICLR.cc/2025/Conference — Submitted to ICLR 2025_

### Official Review · Reviewer_qv4o · 2024-10-29

**Soundness:** 2
**Presentation:** 1
**Contribution:** 2
**Rating:** 3
**Confidence:** 4

**Summary:**

The paper introduces Ego-Foresight, a self-supervised representation learning approach for reinforcement learning (RL), inspired by human learning processes. It focuses on disentangling representations of the agent and its environment by predicting future images using an initial scene representation combined with the robot's proprioception. This scene representation is derived from a set of context images and is constrained to capture only information relevant across the sequence through an additional regularization term. The learned representation is coupled with a jointly trained RL agent, where only the critic's gradients influence the representation. Ego-Foresight is evaluated on tasks from the DeepMind Control (DMC) Suite, two tasks from the Distracting Control Suite, and ten Meta-World tasks.

**Strengths:**

The work is clear, easy to understand, and well-motivated, drawing inspiration from concepts of human learning, which lends intuitive appeal to the proposed methodology. The resulting method is simple and easy to implement.  Furthermore, the work clearly states all used hyper-parameters, making it easy to reproduce.

**Weaknesses:**

Unfortunately, the paper lacks both qualitative and quantitative comparisons with related self-supervised representation learning methods in RL [e.g. 1, 2,3]. Many existing approaches achieve comparable or better results without relying on proprioception, and those that do incorporate it are not referenced and compared to [e.g. 4,5,6] . While state-of-the-art performance is not a strict requirement, fair evaluation and transparent reporting are essential. Notably, the paper overlooks a significant line of research aimed at disentangling agents from environments through bi-simulation metrics [e.g. 7] or explicitly factorized representations [8, 9]. Although these approaches are motivated from a probabilistic perspective, rather than human-centered, I believe they ultimately share the same motivation and are thus highly relevant, and Ego-Foresight should be contextualized accordingly.

Additionally, compared to recent work in this area (e.g. those mentioned above), Ego-Foresight is evaluated on a limited set of relative tasks. The evaluation on the distracting control suite, in particular, could be much more extensive, as the method's emphasis on disentangling agents from environments suggests it should perform well in these scenarios.

A further major concern is the lack of rigor in the statistical analysis. In reinforcement learning, using only 3 or 5 seeds is generally considered inadequate, and standard deviation alone does not sufficiently capture the uncertainty in the results [e.g. 10].

Given the lack of these comparisons to related work, insufficient statistical rigor, and limited evaluation, I believe the paper currently does not meet the bar for acceptance.

[1] Masked World Models for Visual Control, Seo et al 2022

[2] RePo: Resilient Model-Based Reinforcement Learning by Regularizing Posterior Predictability, Zhu et al 2023,

[3] TD-MPC2: Scalable, Robust World Models for Continuous Control, Hansen et al 2024


[4] Robust robotic control from pixels using contrastive recurrent state-space model, Srivastava at al 2021

[5] Mastering diverse domains through world model, Hafner et al 2023

[6] Combining Reconstruction and Contrastive Methods for Multimodal Representations in RL, Becker et al 2024

[7] Learning Invariant Representations for Reinforcement Learning without Reconstruction, Zhang et al 2020

[8] Learning Task Informed Abstractions, Fu et al 2021

[9] Denoised MDPs: Learning World Models Better Than the World Itself, Wang et al 2022

[10] Deep Reinforcement Learning at the Edge of the Statistical Precipice, Agarwal et al, 2021

**Questions:**

- The Distracting Control Suite provides 3 difficulty levels, which one was used for the experiments?

- It is not entirely clear to me how the representation is formed when collecting data or evaluating the agent in the environment. Is the context recomputed after each step using the new image or is there a way of updating it?

---

> ### Author Response · Authors · 2024-11-16
>
> Thank for your comprehensive review, which will help us improve our work.
> We reply to some points that were raised by multiple reviewers in the Official Comment section at the top. We also updated the PDF of the paper with new results.
>
> We will try to provide further detail on the Weaknesses and Questions raised.
>
> - [**Additional baselines**] To improve the comparison against state-of-the-art we update from Dreamer-v2 to Dreamer-v3 [5] on DMControl and add Dreamer-v3 results for Meta-World. We also add TD-MPC [3] results on both DMControl and Meta-World.
>
> - [**Proprioception inputs**] We refer to the Official Comment at the top, where we discuss the use of action/proprioception as input by the other baselines.
> - [**Other lines of research**] We add references to these additional lines of research [7, 8, 9] in the Related work section.
> - [**Number of seeds**] We refer to the Official Comment at the top.
> - [**Q1**]: We use 1 task of the easy level and 3 tasks from the medium level.
> - [**Q2**]: When acting in the environment, the representation is obtained at each time-step using the C most recent steps of RGB frames and proprioceptive states. It is only when the Encoders and Decoder are updated that future proprioceptive states are also used as input.
>
> We thank you again for your review.

---

> > ### Comment · Reviewer_qv4o · 2024-11-25
> >
> > I thank the authors for the clarifications and acknowledge the inclusion of some additional baselines and related works. However, given the discussion regarding proprioception (cf. my answer in the main discussion) and the still insufficient number of seeds (cf. below), I remain with my initial assessment.
> >
> > **Regarding seeds:**
> > Using as few as 3-5 seeds per environment can be ok if a lot of environments are considered and statements are made based on results averaged over the entire suite (c.f. TD-MPC2 and Dreamer-v3, however, this also needs to be done with care). While I can personally relate to the "insufficient compute" argument it unfortunately does not excuse a lack of statistical rigor.

---

### Official Review · Reviewer_C5NB · 2024-11-01

**Soundness:** 2
**Presentation:** 2
**Contribution:** 2
**Rating:** 5
**Confidence:** 3

**Summary:**

This work introduces an approach to improve the sample efficiency of reinforcement learning (RL) by leveraging self-supervised learning to disentangle agent and environment dynamics. Inspired by human motor prediction, the proposed method enables agents to predict future visual states, allowing them to focus on learning task-relevant visuomotor features without the need for supervisory signals. Tested on various simulated environments, including robotic manipulation tasks, the method demonstrates improved performance and efficiency compared to baseline models.

**Strengths:**

1. The idea of using self-supervised agent-environment disentanglement through visuomotor prediction is a fresh and promising approach for improving reinforcement learning (RL) efficiency.

**Weaknesses:**

Weakness & Questions

- It seems obvious that using proprioceptive states provides additional information compared to using only images, so the performance should naturally be better. Therefore, to show the superiority of the proposed method, it would be better to validate it in more complex manipulation environments, involving multiple objects or more intricate interactions.
- Dreamer also predicts future images even without proprioception, just through imagination, and achieve comparable results. Then, why do we need to use this method, additionally preparing datasets paired with proprioceptive states.
- In the training phase, actions are not included in the predictions. When doing RL, the agent is likely to encounter unseen states—won’t this break the representation in such cases?
- The performance doesn’t seem particularly strong. That is, it seems that the proposed method mostly achieves comparable results and does not outperform. How should I interpret these results?
- Why do the baselines differ for each benchmark environment?
- Why did you choose LSTM? Why not use other recent models like Transformer or state space model such as Mamba?
- How does this compare with TD-MPC2[1], one of the SOTA model-based image RL methods?
- What exactly makes this suitable for real-world applications?

[1] Hansen, Nicklas, Hao Su, and Xiaolong Wang. "Td-mpc2: Scalable, robust world models for continuous control." arXiv preprint arXiv:2310.16828 (2023).

**Questions:**

Refer to the Weakness section.

---

> ### Author Response · Authors · 2024-11-16
>
> Thank for your comprehensive review, which will help us improve our work.
> We reply to some points that were raised by multiple reviewers in the Official Comment section at the top. We also updated the PDF of the paper with new results.
>
> We will try to provide further detail on the Weaknesses (W) and Questions (Q) raised.
>
> - **[W1 and W2 (proprioceptive states)]**: we refer to the Official Comment at the top, where we discuss the use of actions/proprioception as input and how Dreamer also makes use of proprioceptive inputs. Additionally, in our choice of benchmarks we sought to include both locomotion/physical control tasks and object interactions in Meta-World. For example, the Hammer and Box Close tasks require complex and precise object manipulation.
>
> - **[W3]**: We may have misunderstood this point. We use future actions/proprioception as input for learning the representations. However, predictions are only future RGB images. To guarantee that most states are observed, we use the babbling stage to cover as many agent body configurations as possible. The model learns the mapping between proprioceptive state and observation of self-configuration, and can generalize to new positions, as demonstrated in the infinity experiment in the appendix.
>
> - **[W4]**: If we look at SEAR as an Oracle - since it uses supervision to obtain the perfect disentanglement between agent and environment - then we perform very close to the oracle. This point could be more explicit in the paper. When compared to the model-based methods we indeed underperform but it is worth it to point out that these approaches using a totally different method with much higher parameter count. Our goal in this work was to demonstrate that self-supervised agent-awareness can improve results of existing methods such as DrQ-v2, and beating the SOTA is out of the scope. It is possible that if we augmented the model-based approaches with agent-awareness their results would also improve.
>
> - **[W5, W7]**: We didn't have the computational resources necessary to obtain results for all the baselines on multiple seeds for all the benchmarks. Nevertheless, we now add results for Dreamer-v3 instead of Dreamer-v2 on both DMControl and Meta-World Benchmarks. We also add a new baseline TD-MPC2 on DMControl and Meta-World. These results were obtained from scores made available online the the authors.
>
> - **[W6 (use of LSTM)]**: We verified that an LSTM was enough to achieve very good predictions on the benchmarks we tested. Because the focus is to predict only the agent while ignoring the rest of the environment, having a model with limited capacity is actually important, otherwise it will also learn the dynamics of other moving bodies, limiting the disentanglement ability of the approach. The LSTM is enough to learn agent dynamics but not so powerful as to predict external dynamics.
>
> - **[W8 (real world applications)]**: For future RL models to be directly trained on real robots, sample efficiency is of paramount importance, more so than wall-clock efficiency. In these applications, due to the speed of the robot, the bottleneck is in the amount of training episodes that can be obtained rather than in the wall clock efficiency of the method.
>
> We thank again you for you review!

---

> > ### Comment · Reviewer_C5NB · 2024-11-25
> >
> > Thank you for your effort in addressing the concerns raised in the reviews. I have carefully considered your responses, as well as the perspectives shared by other reviewers. However, my primary concerns remain unresolved, and my assessment of the work has not changed.

---

### Official Review · Reviewer_3Djm · 2024-11-04

**Soundness:** 1
**Presentation:** 3
**Contribution:** 2
**Rating:** 3
**Confidence:** 4

**Summary:**

This paper presents Ego-Foresight, a self-supervised representation learning method for model-free reinforcement learning (RL). The proposed method is an auxiliary objective that can be implemented on top of common off-policy algorithms like DDPG / DrQ-v2 as in this work, and the representation (encoder) shared between actor-critic and auxiliary prediction head (decoder) is optimized end-to-end using a combination of critic loss (TD-learning) and auxiliary objective (Ego-Foresight; EF). The proposed auxiliary task is to predict future image observations conditioned on a sliding window of recent image observations + a sequence of future proprioceptive states, and as such, the proposed method assumes access to such proprioceptive states from the environment. This is a reasonable assumption in e.g. robotics applications, where joint positions can easily be read with acceptable precision on most robotic manipulators and locomotive robots in the real world. Experiments are conducted on 4 tasks from DMControl, 2 tasks from Distracting Control Suite, and 10 tasks from Meta-World.

**Strengths:**

- The paper is generally well written and easy to follow. The introduction clearly articulates the motivation for the proposed self-supervised task. I believe that the paper is self-contained and provides sufficient background information for unfamiliar readers to appreciate the technical contributions.
- The idea of modeling future image observations solely from proprioceptive information is interesting and likely to work well in problem settings where there is minimal (relevant) visual information external to the agent itself (i.e., object poses are relatively consistent between time steps).
- There is sufficient discussion of limitations in Section 5 (Discussion and Future Work). I appreciate that the authors clearly state limitations that are technical in nature and relevant to the proposed method (not simply regurgitating limitations of visual RL in general).

**Weaknesses:**

- I believe that the experimental setup is flawed. While the argument that proprioceptive information generally is available to an agent in the real world is valid, I don't believe that the chosen benchmarks are appropriate for the point that the authors are trying to make. DMControl / Distracting Control Suite are visual RL benchmarks in which agents usually only have access to raw RGB inputs, so extending DrQ-v2 with the proposed auxiliary task (and thus additional "privileged" proprioceptive information) and comparing this method against vanilla DrQ-v2 and Dreamer-V2 without access to this information is inherently unfair. Additionally, all four DMControl tasks that the authors consider require no object interaction (predominantly locomotion) and can thus easily be solved solely with proprioceptive information (no vision). I therefore cannot tell whether the improvements shown in Figure 4 and 5 (which is very minimal to begin with) are due to privileged observations or the proposed auxiliary objective. I strongly suspect that it is the former.
- A key motivation for the proposed method appears to be "its ability to improve efficiency and performance in different tasks while making strides towards real-world RL applications, by removing the need for costly supervisory signals". How exactly does the method achieve this goal? The method uses the same source of supervision as the algorithm that the methods builds upon (DrQ-v2), namely environment rewards, but then additionally also uses proprioceptive information. If the authors mean to convey that other representation learning methods (different from DrQ-v2) use costly supervisory signals then I would expect a comparison to more such methods.
- There is no discussion or comparison of wall-time between methods. Presumably, adding an auxiliary objective that decodes visual observations up to 40 time steps into the future would be quite computationally expensive (which the authors acknowledge in L485, so I believe it is necessary to report numbers on that.
- Another significant limitation of the method is that it assumes that there is little to no change in the environment between time steps since otherwise the proprioceptive information will not be sufficient to reconstruct relevant content in the future RGB images. The authors do disclose this in the paper itself which I appreciate, but I feel like this significantly limits its practicality.
- The paper compares to Dreamer-V2 (2020), while Dreamer-V3 (2023) has been available for nearly 2 years at this point. I would expect the authors to compare to the best methods available (including Dreamer-V3 as well as other recent methods) especially considering the fact that many such methods (including Dreamer-V3) have publicly available results for download (which is how the authors obtained the DrQ-v2 and Dreamer-V2 numbers according to L323).

**Questions:**

- "Environment steps represent the number of times the environment is updated according to an action which, because we use an action repeat of 2, is always double the number of actions taken by the actor. Environment updates incur a significant computational cost, therefore being the preferred way of reporting sample-efficiency in the literature (Yarats et al. (2021a), Hafner et al. (2020))." Can the authors please clarify what the mean by this? It is possible that I misunderstood the message here. Action repeat of 2 is usually used to make control and TD-learning easier by artificially decreasing the control frequency of the agent, not because environment updates are costly (which will always happen regardless of the action repeat used).

---

> ### Author Response · Authors · 2024-11-16
>
> We thank you for your thorough review and for contributing to the improvement of our work!
> We reply to some points that were raised by multiple reviewers in the Official Comment section at the top. We also updated the PDF of the paper with new results.
>
> We'll try to provide further detail on the Weaknesses (W) and Questions (Q) raised.
>
> - **[W1 (choice of benchmarks)]**: While it is true that DMControl are visual RL benchmarks, it is not uncommon for this benchmark to also be used by approaches that also make use of actions/proprioception as input. This is for example the case of Dreamer-v3 and TD-MPC2, which as discussed in the official comment at the top, also use proprioceptive inputs and are tested with the DMControl benchmark. To provide experiments with object interaction, we used the Meta-World benchmark, where tasks such as Hammer or Box-Close require complex and precise interactions.
>
> - **[W2 (removing supervision)]**: the most direct baseline to our method is SEAR, which extends DrQ-v2 with agent-environment awareness by relying on the availability of supervisory masks of the agent. Our method similarly extends DrQ-v2 with agent-environment awareness but removes the need for supervision. SEAR can therefore be seen as an Oracle model, when compared to our method. We tried to better clarify this point in the paper. We provide comparisons to SEAR on DMControl, Distracting DMControl and Meta-World.
>
> - **[W3 (wall clock)]**: It is true that our method is less efficient in terms of wall-clock time when compared to DrQ-v2. When compared to SEAR it has similar wall-clock requirement. However, we'd like to point out that if future work wants to train RL models directly on real robots, sample efficiency is more important than wall-clock efficiency, as the speed with which the robot moves (which is much slower than in simulation) represents the true bottleneck during each training episode. Hence, the less training episodes the better. Nevertheless, we added this point about wall-clock efficiency to the limitations discussion at the end of the paper.
>
> - **[W4 (change in environment)]**: While it is true that we assume that there is little change in the environment, we believe this requirement might not be a hard requirement since the model can only predict the dynamics that are determined by the future actions/proprioception of the agent. For example, if a person was moving in the background, this movement wouldn't be predictable from the agent's future actions/proprioception. Only the agent's own movement would be correctly predicted. Hence, it would still be possible for the model to disentangle agent and environment by learning what is predictable from it's own proprioception and what is not. In future work we intend to add more of these complex scenarios with motion that is not generated by the agent.
>
> - **[W5]**: we add new Dreamer-v3 results on both DMControl and Meta-World benchmarks. We also add TD-MPC2 results. See the updated PDF.
>
> - **[Q1]**: We have reformulated this sentence to make it more clear in the paper. We don’t mean that we use action repeat of 2 because environment updates are costly. We mean that because we use action repeat 2, the number of actor-steps taken is always half the number of environment steps. Then, we use environment-steps in our axis because these represent the computational cost better than the actor-steps.
>
> Once again, thank you for your review!

---

> > ### Comment · Reviewer_3Djm · 2024-11-28
> > **Thank you**
> >
> > Thank you for responding to my comments and for adding the new results. As discussed in the thread under your general comment, I believe that my main concerns still remain. I thus choose to maintain my score of 3 and recommend rejection. I encourage the authors to take all of the reviewer feedback into account when revising the manuscript for future submission, as there seems to be consensus amongst reviewers. I do think that the general idea and technical contributions are interesting and worth pursuing, but the paper currently has significant flaws that need to be addressed.

---

### Official Review · Reviewer_xayZ · 2024-11-04

**Soundness:** 1
**Presentation:** 3
**Contribution:** 1
**Rating:** 3
**Confidence:** 4

**Summary:**

In this paper, it proposes a self-supervised method, Ego-Foresight (EF), which aims to disentangle agent and environment representations in reinforcement learning (RL) tasks. EF integrates into a model-free RL framework, leveraging agent-centric visuomotor prediction to enhance learning efficiency. The approach is tested in simulated robotic manipulation and locomotion tasks, where it demonstrates some improvements in sample efficiency.

**Strengths:**

The paper presentation is clear, making the reader very easy to follow the proposed learning objectives and architectures.

**Weaknesses:**

1. **Limited Technical Novelty**: Although the authors propose EF as a method for disentangling agent and environment states, the approach appears primarily as next-state prediction for proprioceptive and visual observations. There are no novel technical contribution in model design or learning objectives that set this method apart from existing model-based RL approaches, such as TD-MPC2 [1] or Dreamer-V3 [2], which could similarly incorporate proprioceptive inputs.

2. **Limited Practical Justification**: The paper lacks a compelling argument for EF's applicability in noisy real-world scenarios, where complex background dynamics could compromise the self-supervised model's performance.

3. **Baseline and Comparison Limitations**: The paper’s selection of baselines is narrow and omits recent, relevant advancements in visual motor control on top of DrQ-v2. See Questions for suggestions on alternative baselines that would provide a more rigorous comparison.

4. **Incomplete Baseline Coverage**: For example, Dreamer is only compared in DMC and omitted from Meta-World experiments, resulting in an incomplete analysis across the evaluated domains.

5. **Inconsistent Random Seeds**: The study's use of random seeds is inconsistent and limited, with only 3 seeds for EF in the DMC tasks, which weakens the reliability of performance claims. Such limitations are attributed to time constraints, but this inconsistency detracts from the scientific rigor of the evaluation.

**Questions:**

1. Why does the paper only compare with Dreamer-v2 but not the more recent Dreamer-v3?
2. How does the proposed method compare with other recent model-free and model-based algorithms, such as TD-MPC2, ALIX, TACO, and DRM, which have shown promise in visual motor control tasks?

### References
1. Nicklas et al. *TD-MPC2: Scalable, Robust World Models for Continuous Control*, ICLR 2024.
2. Hafner et al. *Mastering Diverse Domains through World Models*, arXiv Preprint.
3. Cetin, et al. *Stabilizing Off-Policy Deep Reinforcement Learning from Pixels*, ICML 2022.
4. Zheng et al. *TACO: Temporal Latent Action-Driven Contrastive Loss for Visual Reinforcement Learning*, NeurIPS 2023.
5. Xu et al. *DrM: Mastering Visual Reinforcement Learning through Dormant Ratio Minimization*, ICLR 2024.

---

### Author Response · Authors · 2024-12-04

We thank the reviewers for giving their time for the review of our paper and for providing detailed feedback, which we'll take into consideration in trying to improve our future work.

---

### Meta-Review · Area_Chair_Hq2C · 2024-12-24

**Metareview:**

The paper presents a self-supervised approach to disentangle agent-environment information by predicting future visual observations based on proprioceptive states. While the core idea is promising and clearly presented, reviewers identified several critical limitations that led to its rejection.

The primary concerns centered on experimental design and evaluation methodology. A key issue was the unfair advantage gained by incorporating proprioceptive data with visual information in standard image-only RL benchmarks like DMControl, putting baselines without access to joint states at a disadvantage. Additionally, the performance improvements were deemed underwhelming given the extra input information available. The reviewers also noted incomplete comparisons with recent unsupervised methods for learning robust or disentangled representations that use fewer assumptions. These concerns were compounded by statistical weaknesses - limited experimental seeds and missing baseline comparisons - which cast doubt on the significance of the reported results. Given these methodological issues, the reviewers concluded that while the underlying concept had merit, the current implementation and evaluation fell short of the publication threshold.

**Additional Comments On Reviewer Discussion:**

In discussing specific critiques, reviewers highlighted a fundamental issue with the experimental setup - the use of proprioceptive inputs in DMControl tasks potentially masked the true impact of the proposed self-supervised objective, since these tasks can often be solved using proprioceptive information alone. While the authors attempted to address concerns by including comparisons with Dreamer-v3 and TD-MPC2, these additions were deemed insufficient to establish the method's comparative advantages.

The reviewers emphasized two key methodological weaknesses: insufficient statistical validation due to limited seeds and incomplete task averaging, and inadequate baseline comparisons that failed to convincingly demonstrate the method's advantages. Though they acknowledged the promising concept of agent-environment disentanglement, they concluded that acceptance would require substantially expanded experiments - particularly in more challenging domains like complex manipulation or external dynamics - along with more comprehensive baseline comparisons to validate the method's contributions.

---

### Decision · Program_Chairs · 2025-01-22

Reject